# The Adsorption Behavior of Gas Molecules on Mn/N- and Mn-Doped Graphene

**DOI:** 10.3390/nano14161353

**Published:** 2024-08-15

**Authors:** Tingyue Xie, Cuifeng Tian, Ping Wang, Guozheng Zhao

**Affiliations:** 1School of Physical and Electronics Science, Shanxi Datong University, Datong 037009, China; tingyuexie@126.com (T.X.); cftian_050@sxdtdx.edu.cn (C.T.); 2Key Laboratory of Magnetic Molecules & Magnetic Information Materials, Ministry of Education, School of Chemistry and Material Science, Shanxi Normal University, Taiyuan 030031, China; zhaoguozheng@sxnu.edu.cn

**Keywords:** density functional theory (DFT), graphene, transition metal, gas adsorption

## Abstract

By using density functional theory (DFT), the adsorption behavior of gas molecules on defective graphene doped with manganese and nitrogen were investigated. The geometric structure, electronic structure, and magnetic properties of two substrates were calculated and the sensing mechanism was also analyzed. The results indicate that the MnSV-GP and MnN_3_-GP have stronger structural stability, in which Mn atoms and their coordination atoms will become the adsorption point for five gas molecules (CH_2_O, CO, N_2_O, SO_2_, and NH_3_), respectively. Moreover, at room temperature (298 K), the recovery time of the MnSV-GP sensor for N_2_O gas molecules is approximately 1.1 s. Therefore, it can be concluded that the MnSV-GP matrix as a magnetic gas sensor has a promising potential for detecting N_2_O. These results also provide a new pathway for the potential application of Mn-doped graphene in the field of gas sensors.

## 1. Introduction

In recent years, graphene has become popular research topic and attracted considerable interest in many fields [1,2], such as nanoelectronic devices, solar cell technology, liquid crystal electronic devices, and gas sensors [3,4,5,6,7,8]. Owing to the single-layer two-dimensional honeycomb lattice stacked by sp^2^ hybridized carbon, graphene has excellent physical and electrochemical properties, such as zero band gap, massless Dirac particles, anomalous quantum Hall effect, high electron mobility, excellent electrical conductivity, high mechanical strength, and large surface-to-volume ratio [2,9]. While the carrier concentration be changed by the adsorption of gas molecules, indicating that graphene can be used as sensitive sensors [5,10]. However, the interaction between most gas molecules and graphene is limited to physical adsorption [11,12,13], which is not strong. In order to improve the adsorption of graphene, people have attempted to modify graphene or fabricate hole-matrixed graphene [14]. Reference [15] demonstrates that modified graphene can enhance selectivity and sensitivity towards certain specific gas molecules. Therefore, it is very important to find suitable methods to improve the sensitivity and selectivity of graphene.

Single-atom catalysts (SACs), which has isolated metal atom singlely dispersed on support, can greatly improve the distribution of active sites, catalyst stability, reaction selectivity, and gas sensitivity [16,17,18]. The combination of carbon-based sensors and single-metal-atom catalysts can bring particular electronic structures and enhance gas-sensing capabilities. Experimentally, it has been proved that transition-metal-doped graphene or transition-metal-and-nitrogen co-doped graphene has significantly influenced electronic structure and its electrocatalytic activities [19,20]. Recently, the adsorption behavior of substrates of various graphene matrices doped with transition metal (TM) atoms, such as TM-C or TM-N-C has been researched [21,22,23,24,25]. The experimental results have confirmed that Pd-doped graphene can significantly enhance the selectivity and sensitivity for NH_3_ gas molecules [26]. Some theoretical studies have shown that doped graphene can enhance the adsorption of small gas molecules (H_2_, CH_2_O, CO, NO, and SO_2_) compared with pure graphene [27,28,29]. In addition, when some small gas molecules adsorbed on doped metal graphene, the spin state of the Fermi level and the magnetism of the substrate were simultaneously changed [30,31,32,33]. Considering the lower binding energy and the stronger interaction between transition metal atoms and carbon (nitrogen) structures, the adsorptivity and magnetism of the substrate was greatly affected. So, it is important to understand this interaction mechanism.

In recent research, the density functional theory has been used to calculate and study various gas-sensing materials, explain the characteristics of gas adsorption, and design new gas-sensing materials. The stable geometric structure, electronic structure, magnetic properties, and catalytic activity of the doped defective graphene with Mn atom in single vacancies (SV) or among three nitrogen atoms (N_3_) have been studied [11,20,34,35,36,37]. However, so far, the systematic studies on the adsorption of gases (CH_2_O, CO, N_2_O, SO_2_, and NH_3_) on MnSV-GP or MnN_3_-GP have not been reported. Here, the above molecules as the adsorption probes, the geometric structure, electronic structure, and magnetic properties of two substrates (MnSV-GP and MnN_3_-GP) were investigated, respectively. Based on density functional theory (DFT), the gas-sensing mechanism is analyzed through density of states theory (DOS), charge transfer, spin density, and charge density difference. Finally, to evaluate the repeatability and the utilization rate of gas sensor, the recovery time of gas desorption was discussed.

## 2. Computational Details and Methods

All calculations are conducted by using the Vienna ab initio simulation package (VASP), which is based on density functional theory (DFT) with a van der Waals (vdW) correction for the spin-polarized surface system [38,39]. The electron exchange energy and core-electron interactions were described by the Pedrew Burke Ernzerhof of generalized gradient approximation (GGA) and projection augmented wave (PAW) methods, respectively [40,41]. The maximum cutoff energy of PAW was set to 500 eV. In the process of geometric optimization, all atomic positions were fully relaxed until the forces were smaller than 0.01 eV/Å. The total energy convergence standard for electronic self-consistent steps is less than 1 × 10^−6^ eV. According to references [42,43], a 4 × 4 × 1 supercell model for simulating two-dimensional hexagonal graphene substrates was applied. The calculations of various free gas molecules were performed by using a crystal cell of 15 × 15 × 15 Å^3^. The k-point mesh of 15 × 15 × 1 is used to calculate the density of state (DOS) and 7 × 7 × 1 Γ-center mesh for other calculations. When the energy change is less than 10 meV/atom, the k-point grid test is finished. In this work, the spin-polarized DFT was used to calculate magnetic properties. It has been proven that the method is reasonable for calculating transition-metal-doped graphene [44,45]. In order to further accurately describe the van der Waals (vdW) interaction, we adopted the DFT-D2 method, which has been successfully applied to the theoretical study of adsorption systems [39]. A vacuum layer of 15 Å along the vertical direction was set to minimize the periodic interaction. A Gaussian smearing method with a width of 0.05 eV was used to improve convergence of the orbital occupation near the Fermi level. In addition, Bader charge analysis was used to evaluate the atomic charge and electron transfer amount in adsorption [46], and the electron density difference was drawn by visualization software VESTA 3 [47].

The adsorption energy of stable configurations (Eads) is expressed as
Ead = (Egas + Esup) − Egas/sup
where Egas, Esup, and Egas/sup denote the total energy of single gas molecules in vacuum, the MnSV-GP or MnN_3_-GP support, and the gas adsorbed on the support, respectively. Based on this definition, a positive value shows that the adsorbed system is more stable and the gas adsorption is exothermic.

## 3. Results and Discussion

### 3.1. Stability and Structure for MnSV-GP and MnN_3_-GP

In order to study the stability of MnSV-GP and MnN_3_-GP, the structure of the system was studied. The most stable atomic configurations of MnSV-GP and MnN_3_-GP are shown in Figure 1a,b, respectively. Their structural parameters are shown in Table 1. Consistent with previous calculations [48,49,50,51], the bond lengths of Mn-C (1.83 Å) and Mn-N (2.00 Å) in these two structures are both smaller than the covalent radii of Mn atoms and C or N atoms [52], indicating that Mn-C or N atoms form stable covalent bonds. The binding energy is up to −6.12 eV and −4.2 eV for MnSV-GP and MnN_3_-GP, respectively, which is consistent with previous theoretical calculations [48,50]. The value of binding energy is significantly lower than that of cohesive energy of Mn atoms obtained from experiments (−2.92 eV) and theoretical calculations (−3.86 eV) [53]. It shows there is a strong interaction between the Mn dopant and the graphene surface. According to previous studies [29,50,54], the catalytic activity of MnSV-GP and MnN_3_-GP mainly comes from the active region of the doped atoms (Mn with the nearest C and N atoms).

The stability of these two structures can also be explained by the Bader charge transfer on the substrate. As shown in Table 1, Mn transferred significant charges of 0.88 e and 0.96 e to the SV-GP and N_3_-GP layers, respectively, further indicating that it is chemical adsorption. It is due to that there is a difference in the electro negativity of Mn, N, and C in these two structures, which changes the local electro neutrality of the structure. As shown in Figure 1c,d, it can be observed that there is a significant electron overlap between the Mn atom and surrounding atoms, which will keep the Mn atom stable in the vacancy.

### 3.2. Adsorption Structure and Adsorption Energy

Then, five gas molecules as adsorbates were used to probe the reactivity of two substrates, including MnSV-GP and MnN_3_-GP. The optimized structures can be seen in Appendix A. After optimization, the adsorption energy (Figure 2), charge transfer (Table 2), adsorption height between the adsorbed molecule and Mn–graphene nanosheets, and bond length (summarized in Appendix A, respectively) were employed to investigate the change in structure and properties. As shown in Figure 2, it can be seen that the smallest adsorption energy of the MnN_3_-GP for CH_2_O molecule is −2.26 eV, while the largest one is −0.77 eV in the N_2_O/MnSV-GP system. Meanwhile, for MnN_3_-GP, each of the adsorption energies is evidently smaller than that of MnSV-GP. It is indicated that the interaction between the adsorbed gases and MnN_3_-GP is significantly enhanced, and the sensitivity of substrate is improved through replacing C atoms with non-metallic N atoms.

To further illustrate the adsorption activity, the relationship between adsorption energy and bond lengths were investigated. After the five gas molecules’ adsorption, the average bond length of C-Mn stretched by about 1.27%, −0.18%, −0.36%, 0.73%, and −1.64% compared with that before adsorption, respectively, while N-Mn bonds elongated by approximately 2%, −0.5%, 1.67%, 1%, and −0.5% (Appendix A). As showed in Figure 2, it indicated that the smallest adsorption energy corresponds to the longest bond length. The results also showed that the substrate of MnN_3_-GP has higher stability than MnSV-GP. In the same way, we can see that the bond length in the other four molecules is elongated after adsorption on two surfaces of MnSV-GP and MnN_3_-GP, respectively. In addition, in Appendix A, the bond lengths of C-H1 and C-H_2_ are shortened from 1.12 Å to 1.10 Å under the interaction, while the bond lengths of C-O are extended from 1.21 Å to 1.33 Å and 1.37 Å, respectively. These results indicate that the molecular polarity is strengthened and the adsorption of CH_2_O molecule is activated. Generally, the greater the elongation of bond length between atoms, the smaller the adsorption energy is. When the adsorption force increases, the adsorption activity becomes greater.

### 3.3. Electronic Properties and Magnetic Properties

To further illustrate the adsorption mechanism, we calculated the density of states (DOSs) and projected density of states (PDOSs) of the five gas molecules adsorbed on MnSV-GP and MnN_3_-GP, respectively, as shown in Figure 3 and Figure 4. In Figure 3a–d, by comparing the total density of states (TDOSs) before and after four gas molecules’ (CH_2_O, CO, NO_2_, and SO_2_) adsorption, it can be observed that the TDOS curve shifts towards higher energy, especially near the Fermi level, resulting in partial filling of the valence band (VBM). However, the curve of DOS shifts to lower energy after NH_3_ adsorption (Figure 3e). By analyzing the PDOS curves (in Figure 3), it can be seen that the energy levels of conduction band are mainly the contributions of the 3d electronic states of Mn atoms. Moreover, the DOS near the Fermi level indicates that there is an overlap between the 3d-orbital of Mn atoms and the p-orbital of gas molecules for MnSV-GP and MnN_3_-GP catalysts. As shown in Figure 4a–d, for the MnN_3_-GP, strongly hybridized states between the d-orbitals of Mn atoms and the p-orbitals of gas molecules (CH_2_O, CO, NO_2_, and SO_2_) can be observed near the Fermi energy level. This electronic interaction leads to the formation of bonds between the gas and the Mn, resulting in lower adsorption energies.

Comparing the peak value at Fermi level in Figure 3 with the one in Figure 4, the peak values of the five gas molecules adsorbed on the MnSV-GP surface at Fermi level are 0.1 states/eV, −0.09 states/eV, 0.01 states/eV, 0.02 states/eV, and 0.06 states/eV, while on the MnN_3_-GP surface, they are 6.74 states/eV, −7.81 states/eV, 3.76 states/eV, 8.07 states/eV, and 1.38 states/eV, respectively (local enlarged drawing in Appendix A). The electron concentration at the Fermi level has increased, indicating that N atom substitution has a significant impact on gas adsorption. The MnN_3_-GP system improved the adsorption sensitivity of the five gas molecules.

In order to further investigate the reasons of adsorption sensitivity, we analyzed the electronic structure by the charge transferring and the charge difference density (CDD). According to the Bader charge calculation results (Table 2), it can be seen that the CH_2_O, CO, NO_2_, and SO_2_ molecules will gain electrons, while the NH_3_ molecule will lose electrons in the gas adsorption process. At the same time, it revealed that the distributions of electrons in the gas/MnN_3_-GP are significantly higher than those in the gas/MnSV-GP. On the other hand, according the charge difference density, the CH_2_O, CO, NO_2_, and SO_2_ molecules (Figure 5a–d,f–i) are mainly surrounded by yellow, which indicates that these molecules are electron acceptors. In contrast, for the NH_3_ molecule (Figure 5e,j), most charge density is displayed in cyan, demonstrating that the NH3 molecule is an electron donor. As a result, the migration and redistribution of electrons near the surface will form strong bonds, resulting in a higher response.

In order to clarify the magnetic property, we have also calculated the spin density of five gases adsorbed on the MnSV-GP and MnN_3_-GP substrates, respectively (Figure 6). Spin density is the difference in electron density between the spin up and spin down [55]. When the five gas molecules adsorbed on the MnSV-GP substrate, the magnetic moments of the MnSV-GP reduced from 3.00 μB (Table 1) to 1.0 μB (Table 3). For another substrate (MnN_3_-GP), comparing the magnetic moment before adsorption to that after adsorption, the magnetic moment decreased from 5.15 μB (Table 1) to 3.93 μB, 4.27 μB, 3.98 μB, 3.76 μB, and 4.70 μB (Table 3), respectively. The change in magnetic moment indicates the sensitivity of gas adsorption. In Figure 6a–l, the spin mainly concentrates on Mn atoms with spin-up. In the MnN_3_-GP system, as shown in Figure 6h–l, the nearest N and Mn atom is dominated by upward spin states. The spin direction of the CO molecule is consistent with that of Mn atoms (Figure 6i), while the CH_2_O, N_2_O, and SO_2_ molecules have a dominant downward spin state. These are mainly the results of charge transfer from Mn atoms to CH_2_O, CO, NO_2_, and SO_2_ gas molecules or NH_3_ charge transfer to Mn atoms. In addition, in the case of NH_3_ adsorption, the spin density is strongly limited to Mn atoms and their nearest atoms (Figure 6f,l), while other atoms have lower magnetization intensity. Furthermore, in Figure 6, the adsorption of CH_2_O, CO, NO_2_, SO_2_, and NH_3_ gives rise to polarization for the entire system compared with original substrates (Figure 6a,g). Based on the spin density distribution, it is concluded that the magnetism has changed through the adsorption, and the substrates have higher adsorption sensitivity to the five gas molecules.

### 3.4. Recovery Time

After evaluating the adsorption energy and analyzing the charge transfer mechanism, as another essential parameter for gas sensors, the recovery time need to be calculated. The shorter the recovery time is, the better the repeatability and the higher the utilization rate of a gas sensor. According to traditional transition state theory [56], the recovery time of the system can be estimated as
(1)$$\tau =\nu_{0}^{-1}e^{\left(-E_{B}/K_{B}T\right)}$$
where *τ* is the recovery time and *ν*_0_ is the attempt frequency (10^13^ s^−1^), *K_B_* is the Boltzmann constant, *T* is the temperature (*K*), and *E_B_* is the desorption energy, which can be approximated as the adsorption energy [57]. According to this equation, at room temperature (298 K), the recovery times of CH_2_O, CO, N_2_O, SO_2_, and NH_3_ on the MnSV-GP sensor were calculated to be 5.27 × 10^13^ s, 4.6 × 10^9^ s, 1.1 s, 1.3 × 10^18^ s, and 1.2 × 10^5^ s, respectively. The recovery times for MnN_3_-GP were 1.7 × 10^25^ s, 2 × 10^18^ s, 5 × 10^11^ s, 2.3 × 10^24^ s, and 4.4 × 10^8^ s, respectively. At room temperature, the MnSV-GP sensor has a short recovery time (1.1 s) for N_2_O molecules. In this way, the MnSV-GP matrix can be predicted as a promising and reusable magnetic gas sensor for detecting N_2_O, with high selectivity and sensitivity.

## 4. Conclusions

In this study, we have performed density functional theory to study the adsorption of gas molecules, including CH_2_O, CO, N_2_O, SO_2_, and NH_3_, on the MnSV-GP and MnN_3_-GP substrate, respectively. The adsorption geometry, adsorption energy, and electronic and magnetic properties of the adsorption systems have been investigated, and the recovery time has been analyzed according to traditional transition state theory. It was concluded that the MnN_3_-GP support has a remarkable influence on the adsorption characteristics of the five gas molecules. It has also been found that the MnN_3_-GP support possesses higher stability and adsorption activity than the MnSV-GP. In addition, the N_2_O/MnSV-GP system has the lowest recovery time, illustrating that this sensor has excellent selectivity for detecting N_2_O. Besides that, this study on spin density can conclude that the two supports have the lower selectivity for the NH_3_ gas than the other four gas molecules. In general, it is shown that Mn-doped graphene have prominent stability, sensitivity, and selectivity in sensing, making them great candidates for application in the field of gas sensors.

## Figures and Tables

**Figure 1 nanomaterials-14-01353-f001:**
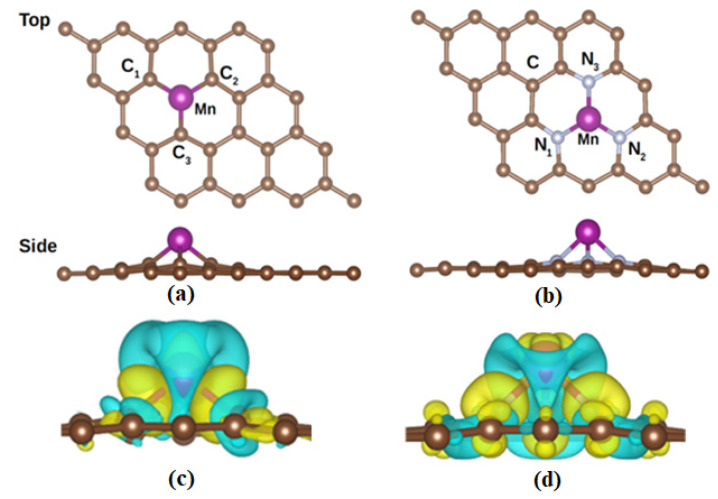
(**a**,**b**) The geometric structure, top and side view, obtained from geometry optimization. (**c**,**d**) Charge density difference in MnSV-GP and MnN_3_-GP, respectively. Yellow contours indicate electron accumulation, and cyan contours indicate electron depletion. And other colors represent the chemical elements.

**Figure 2 nanomaterials-14-01353-f002:**
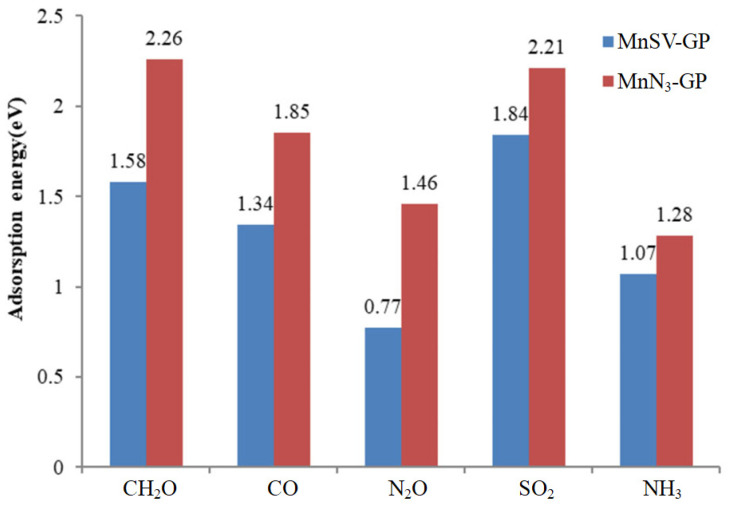
The adsorption energies of five gas molecules adsorbed on two Mn/GN catalysts. Each calculated binding energy is denoted as its absolute value.

**Figure 3 nanomaterials-14-01353-f003:**
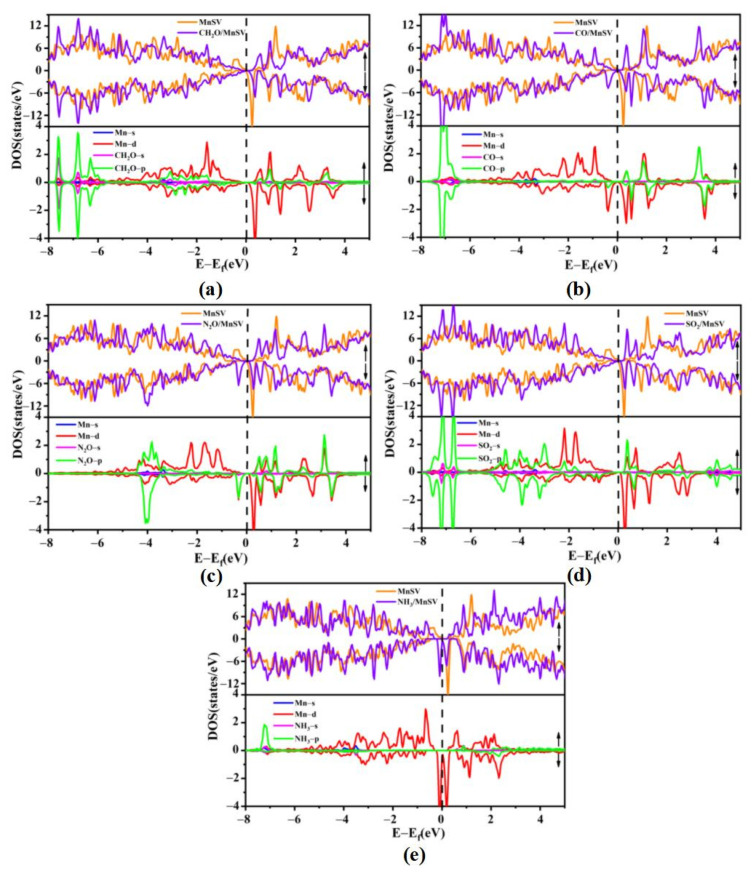
DOS and PDOS distribution: before and after gas molecule adsorption on MnSV-GP (**a**) CH_2_O, (**b**) CO, (**c**) N_2_O, (**d**) SO_2_, and (**e**) NH_3_. Fermi energy is set to zero.

**Figure 4 nanomaterials-14-01353-f004:**
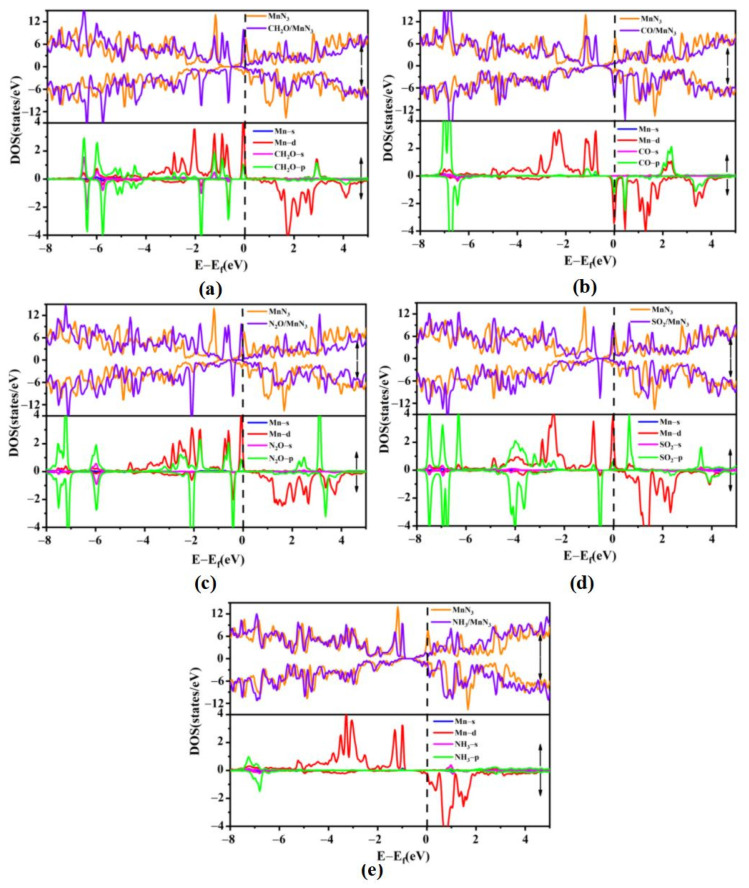
DOS and PDOS distribution: before and after gas molecule adsorption on MnN_3_-GP (**a**) CH_2_O, (**b**) CO, (**c**) N_2_O, (**d**) SO_2_, and (**e**) NH_3_.

**Figure 5 nanomaterials-14-01353-f005:**
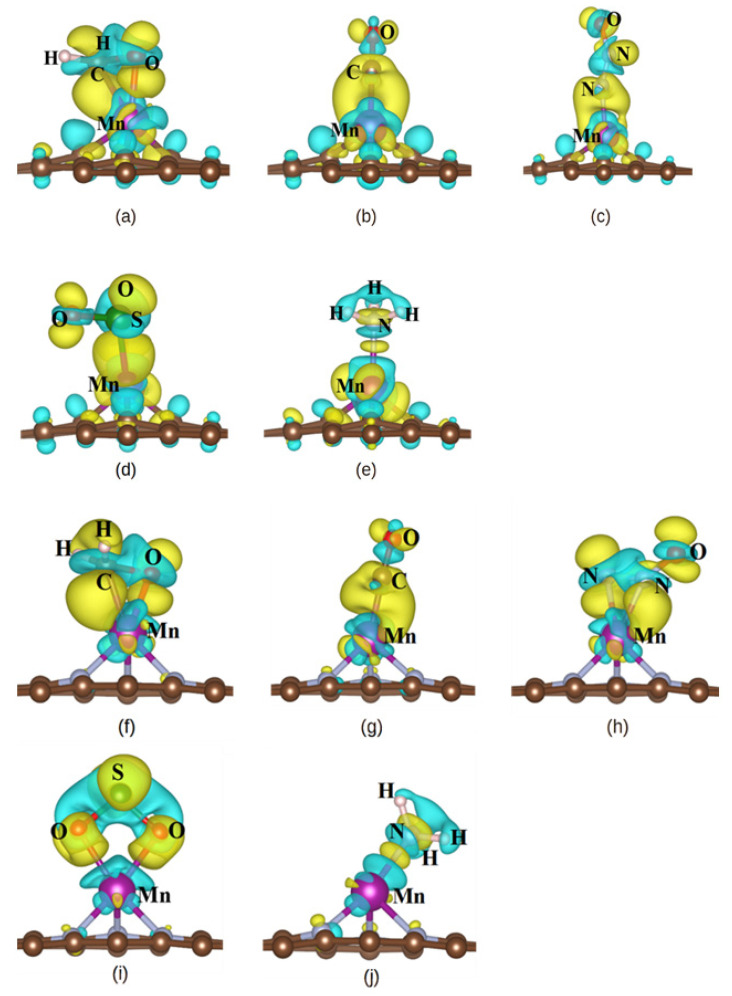
The charge density differences in (**a**–**e**) CH_2_O, CO, N_2_O, SO_2_, and NH_3_ adsorbed on the MnSV-GP and (**f**–**j**) CH_2_O, CO, N_2_O, SO_2_, and NH_3_ adsorbed on the MnN_3_-GP configuration. Yellow contours indicate electron accumulation, and cyan contours indicate electron depletion. And other colors present chemical elements, such as purple color denotes Mn atom. Isosurface value: 0.002 e/Bohr^3^.

**Figure 6 nanomaterials-14-01353-f006:**
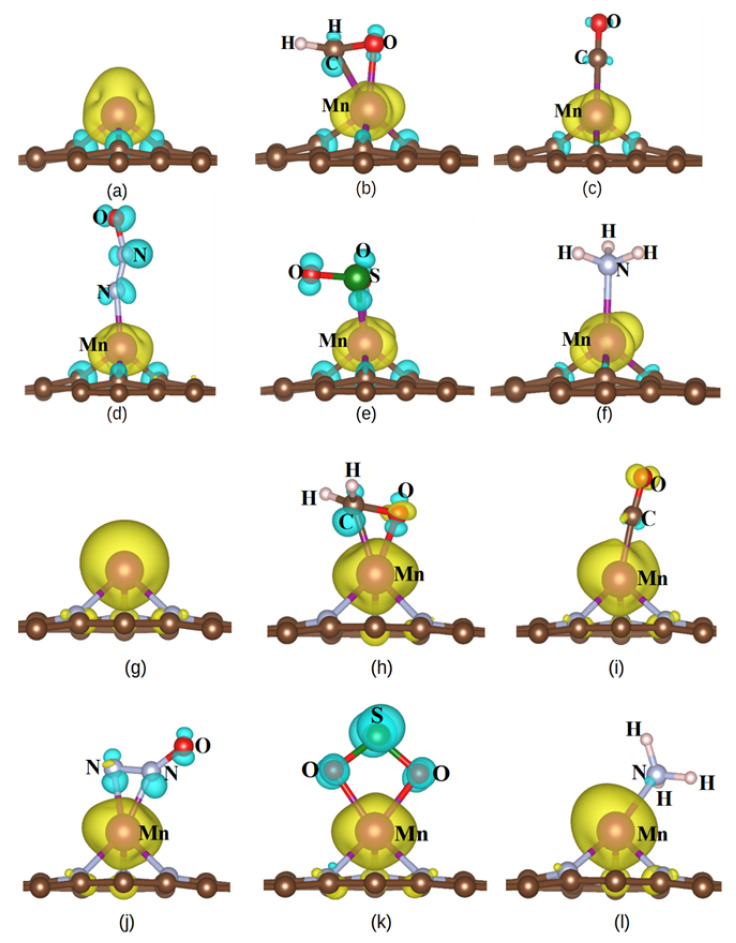
The spin density for two supports and gas-support chemisorption systems. (**a**) The spin density of the MnSV-GP before adsorption; (**b**–**f**) The spin density of the MnSV-GP after adsorption of CH_2_O, CO, N_2_O, SO_2_, and NH_3_, respectively; (**g**) The spin density of the MnN_3_-GP before adstorption; (**h**–**l**) The spin density of the MnN_3_-GP after adsorption of CH_2_O, CO, N_2_O, SO_2_, and NH_3_, respectively. The yellow and cyan areas indicate the positive and negative spin densities respectively; isosurface value: 0.005 e/Bohr^3^. And other colors present chemical elements, such as red and green color represent oxygen and sulfur atom, respectively.

**Table 1 nanomaterials-14-01353-t001:** The structural parameters of MnSV-GP and MnN_3_-GP, including binding energies (*E_b_*), the bond length (*d*) between the manganese atom and the nearest three C or N atoms, the height (*h*) of Mn atoms above the graphene surface, the total magnetic moment (*M*), and the charge transfer (Δ*q*) of Mn atoms on MnSV-GP and MnN_3_-GP.

Substrate	*E_b_* (eV)	*d* (Å)	*h* (Å)	*M* (μ_B_)	Δ*q* (e)
MnSV-GP	−6.12 (−6.39 [48])	1.83 (1.83 [49])	1.40 (1.37 [48])	3.00 (3.00 [49])	0.88 (0.90 [48])
MnN_3_-GP	−4.20 (−4.35 [50])	2.00 (2.00 [51])	1.65 (1.65 [51])	5.15	0.96 (0.97 [51])

**Table 2 nanomaterials-14-01353-t002:** The change in adsorbed gases in valence electron number (the symbols “+” and “−” represent electronic gain and loss).

Gas	Δ*q*-Gas	
	On the MnSV-GP	On the MnN_3_-GP
CH_2_O	+0.41	+0.67
CO	+0.25	+0.37
N_2_O	+0.38	+0.72
SO_2_	+0.49	+0.75
NH_3_	−0.11	−0.09

**Table 3 nanomaterials-14-01353-t003:** The total magnetic moment of the adsorption systems.

Gas	M (μB)	
	On the MnSV-GP	On the MnN_3_-GP
CH_2_O	1.00	3.93
CO	1.00	4.27
N_2_O	1.00	3.98
SO_2_	1.00	3.76
NH_3_	1.00	4.70

## Data Availability

Data are contained within this article.

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
