# Peer review of "The Adsorption Behavior of Gas Molecules on Mn/N- and Mn-Doped Graphene"

_nanomaterials, 2024, doi:10.3390/nano14161353_

Round 1

Reviewer 1 Report

Comments and Suggestions for Authors

This work’s finding is mostly based on the work from Ref. 54, except focus on Mn-SV- and MnN3-GP. The methodology used is reasonable, however it cannot be published in present form. My concerns are following:

1) There has been experimental report in FeN4-GP and MnN4-GP (Nano Lett. 15 (11) (2015) 7408). Is there any experimental reports of MnSV-GP and MnN3-GP? A discussion related to current experimental studies to support the presence of these graphitic system are needed.

2)  The adsorption energy is related to spin state of the individual MnSV- or MnN3-GP system (Table 1) and adsorption system (Table 3). Are these the lowest energy state? Some justification and discussion for the identification of the lowest energy spin state is needed.

3) pg. 2, sect. 2, line 82: DFT-D1: Is it a typo? The Ref. 36 is refer to DFT-D2.

4) pg. 7, line 181-182: “… 0 eV, 0 eV, 7.52 eV, ….” and “9.9 eV, 8.1 eV, …” (pg. 8, line 182) what are these values and where to get those from Fig. 3 & 4.  These values are not consistent with the energy scale shown in Fig. 3 & 4, and the related discussion is confusing. More explanation is needed. 

5) Table 3: Why the total magnetic moment of all the adsorption systems on the MnSv-GP is exactly the same (i.e. 1.0 )?

6) pg. 9, line 237: the assumed attempt frequency (1013 s-1) could be wrong, and cannot lead to those predicted values in recovery time that mentioned in Sect. 3.4.

7) pg. 10, line 246-247: Why at 498K, N2O/MnSV-GP is not discussed since it has the smallest adsorption energy, whereas CO and NH3 is discussed. Is there any reason?  At such a high temperature, a short discussion of thermal stability of adsorption system should be discussed (e.g. Langmuir 2024, 40, 13, 6703–6717). In my opinion, a simple extrapolation to high temperature simply based on “zero-temperature” DFT adsorption energy (Fig. 2) with “frozen-optimized geometry” can be very misleading.

Comments on the Quality of English Language

Appropriate. 

Author Response

Comments and Suggestions for Authors

This work’s finding is mostly based on the work from Ref. 54, except focus on Mn-SV- and MnN3-GP. The methodology used is reasonable, however it cannot be published in present form. My concerns are following:

 Response: Many thanks for the encouraged comments. We have improved below suggestions point to point in the new version.

1) There has been experimental report in FeN4-GP and MnN4-GP (Nano Lett. 15 (11) (2015) 7408). Is there any experimental reports of MnSV-GP and MnN3-GP? A discussion related to current experimental studies to support the presence of these graphitic system are needed.

Response, we really appreciate referee for suggesting the good point to improve the quality of this manuscript. Yes, there some experimental reports of Mn doped graphene (e.g. Chem. Asian J. 2013, 8, 1295 – 1300, now reference 20). We have not found the experimental report on the MnN3-GP. There is a paper “Synergy among manganese, nitrogen and carbon to improve the catalytic activity for oxygen reduction reaction” (Journal of Power Sources 251 (2014) 363-369). Although the elements can match, the structure is different. In the revised manuscript, we added one experimental paper in introduction, ref [19] and changed one reference 34 to 20.

2)  The adsorption energy is related to spin state of the individual MnSV- or MnN3-GP system (Table 1) and adsorption system (Table 3). Are these the lowest energy state? Some justification and discussion for the identification of the lowest energy spin state is needed.

 Response: Yes, these are the lowest energy states for adsorption system. For the adsorption research, through optimizing the geometric structure and electron spin configuration (fixed spin parameters ISPIN=2), the stability configuration was obtained and at same time, the lowest energy was achieved. Identifying the lowest energy of spin state needs to set different spin configuration (e.g. high spin state, low spin state and mix spin state), then compare the energy of different spin configuration, and obtain the lowest energy spin state. In our manuscript, we don’t set different spin configuration, so it is difficult for us to identify the lowest energy of spin state.

3) pg. 2, sect. 2, line 82: DFT-D1: Is it a typo? The Ref. 36 is refer to DFT-D2.

 Response: Thank you for reminding and we have realized the mistake and revised it in the manuscript.

4) pg. 7, line 181-182: “… 0 eV, 0 eV, 7.52 eV, ….” and “9.9 eV, 8.1 eV, …” (pg. 8, line 182) what are these values and where to get those from Fig. 3 & 4.  These values are not consistent with the energy scale shown in Fig. 3 & 4, and the related discussion is confusing. More explanation is needed.

Response: Thanks for your good comments and sorry for the confusion we brought to you. In the revised manuscript, we marked these values with more detailed labeling the unit of DOS. Among these units, such as, states/eV, electrons/eV, and eV(Commun. Theor. Phys. 72 (2020) 035501, as described at page 8, line 8-12), we choose states/eV as the unit of DOS. In addition, many of these values were incorrect, so in the revised supporting information, to ensure measurement data accuracy and clearly, we add Figure S2 which is the enlargement of partial position of Figure 3 and Figure 4.

 5) Table 3: Why the total magnetic moment of all the adsorption systems on the MnSv-GP is exactly the same (i.e. 1.0 )?

Response: In fact, when we look back to the data about five gases adsorption on the MnSV-GP, the magnetic moment is exactly the same value 1.0 μB. It is illustrated that the magnetic moment changed not obviously after gases adsorption on the MnSV-GP support, in contrast, Mn and N co-doped graphene is more sensitivity to gases (Table 3).

 6) pg. 9, line 237: the assumed attempt frequency (1013 s-1) could be wrong, and cannot lead to those predicted values in recovery time that mentioned in Sect. 3.4.

Response: Sorry for the mistake and we have revised it in the manuscript. The assumed attempt frequency is 1013s-1. Similarly, we have made revisions one by one in the text.

7) pg. 10, line 246-247: Why at 498K, N2O/MnSV-GP is not discussed since it has the smallest adsorption energy, whereas CO and NH3 is discussed. Is there any reason?  At such a high temperature, a short discussion of thermal stability of adsorption system should be discussed (e.g. Langmuir 2024, 40, 13, 6703–6717). In my opinion, a simple extrapolation to high temperature simply based on “zero-temperature” DFT adsorption energy (Fig. 2) with “frozen-optimized geometry” can be very misleading.

Response: First, I’ll explain the reason why CO and NH3 are discussed while other not. Among five gases, those with shorter recovery time are showed in manuscript. Second, in the paper (Langmuir 2024, 40, 13, 6703–6717) the interaction energy was calculated by using CP method to correct the energy error even at high temperature. While in our manuscript, we just used the formula of interaction energy (Ead = (Egas + Esup) –Egas/sup). So, in order to ensure accuracy, we decided to delete the result at high temperature. Again, thanks for the reminder avoiding the possible mistake caused by the extrapolation from zero temperature to high temperature.

Reviewer 2 Report

Comments and Suggestions for Authors

Reviewer report on manuscript nanomaterials-3153033

Tingyue Xie   et al.The Adsorption Behavior of Gas Molecules on Mn/N Mn–Doped Graphene

In this work, by using density functional theory (DFT), the adsorption behavior of gas molecules on the defect graphenes embedded by manganese and nitrogen were investigated. The geometric structure, electronic structure, and magnetic properties of two substrates were calculated and the sensing mechanism was also analyzed. The results indicate that the MnSV-GP and MnN3-GP have stronger structural stability, in which Mn atom and their coordination atoms will become adsorption point for five gas molecules (CH2O, CO, N2O, SO2, and NH3), respectively. Moreover, at room temperature (298K), the recovery time of the MnSV-GP sensor for N2O gas molecules is approximately 1.1 seconds. Therefore, it can be concluded that the MnSV-GP matrix as a magnetic gas sensor has a promising potential for detecting N2O.

The manuscript can be accepted after minor revision.  Authors should make the following corrections:

1.      The title of the manuscript should be corrected. I suppose it should be “The Adsorption Behavior of Gas Molecules on Mn/N and Mn–Doped Graphene”.

2.      I recommend Authors to extend the section introduction and add some additional discussion about different detection methods and materials used for detection. Some important up-to-date references in this field are still missed, e.g. [Toward On-Chip Multisensor Arrays for Selective Methanol and Ethanol Detection at Room Temperature: Capitalizing the Graphene Carbonylation. ACS Applied Materials & Interfaces, 2023, 15(23), 28370–28386] and [Guiding graphene derivatization for the on-chip multisensory arrays: From the synthesis to the theoretical background. Advanced Materials Technologies, 2022, 7(7), 2101250], and references there.

3.      More details to the section “2. Computational Details and Methods” should be added.

4.      The obtained results are clearly presented in the manuscript.

5.      The conclusions are supported by the results.

6.      English language is fine.

7.      Typos should be corrected, e.g.:

Line 27, “sp2” should be “sp2

Author Response

Comments and Suggestions for Authors

Reviewer report on manuscript nanomaterials-3153033

Tingyue Xie   et al. “The Adsorption Behavior of Gas Molecules on Mn/N Mn–Doped Graphene” 

In this work, by using density functional theory (DFT), the adsorption behavior of gas molecules on the defect graphenes embedded by manganese and nitrogen were investigated. The geometric structure, electronic structure, and magnetic properties of two substrates were calculated and the sensing mechanism was also analyzed. The results indicate that the MnSV-GP and MnN3-GP have stronger structural stability, in which Mn atom and their coordination atoms will become adsorption point for five gas molecules (CH2O, CO, N2O, SO2, and NH3), respectively. Moreover, at room temperature (298K), the recovery time of the MnSV-GP sensor for N2O gas molecules is approximately 1.1 seconds. Therefore, it can be concluded that the MnSV-GP matrix as a magnetic gas sensor has a promising potential for detecting N2O.

The manuscript can be accepted after minor revision.  Authors should make the following corrections:

 Response: Many thanks for the confirmation and encouraged comments. We have improved below suggestions point to point in the new version.

  1. The title of the manuscript should be corrected. I suppose it should be “The Adsorption Behavior of Gas Molecules on Mn/N and Mn–Doped Graphene”.

Response: Thanks for your good suggestion. In this manuscript, we have replaced the title with “The Adsorption Behavior of Gas Molecules on Mn/N and Mn–Doped Graphene”.

  1. I recommend Authors to extend the section introduction and add some additional discussion about different detection methods and materials used for detection. Some important up-to-date references in this field are still missed, e.g. [Toward On-Chip Multisensor Arrays for Selective Methanol and Ethanol Detection at Room Temperature: Capitalizing the Graphene Carbonylation.ACS Applied Materials & Interfaces, 2023, 15(23), 28370–28386] and [Guiding graphene derivatization for the on-chip multisensory arrays: From the synthesis to the theoretical background. Advanced Materials Technologies, 2022, 7(7), 2101250], and references there.

Response: Referee indeed suggested a very exact point to improve the quality of this work. It is very nice that referee may suggest us the literatures, thus we have cited all the suggested reference. See, ref [8, 14]. Moreover, we added one experimental paper in introduction, ref [19] and changed one reference 34 to 20.

  1. More details to the section “2. Computational Details and Methods” should be added.

Response: Thanks a lot for your comment. We have increased the detailed description in the section “2. Computational Details and Methods”.

  1. The obtained results are clearly presented in the manuscript.

 Response: Thank you for your meaningful evaluation.

  1. The conclusions are supported by the results.

 Response: Many thanks for the confirmation and encouragement.

  1. English language is fine.

 Response: Thanks very much.

  1. Typos should be corrected, e.g.:Line 27, “sp2” should be “sp2”

Response: Sorry for the mistake and we have revised it in the manuscript. Similarly, we have made revisions one by one in the text.

Round 2

Reviewer 1 Report

Comments and Suggestions for Authors

The authors have addressed all the comments from the reviewer. The revised manuscript is better now and can be published as it is.